# The Challenge: Equal Availability to Palliative Care According to Individual Need Regardless of Age, Diagnosis, Geographical Location, and Care Level

**DOI:** 10.3390/ijerph19074229

**Published:** 2022-04-01

**Authors:** Bertil Axelsson

**Affiliations:** Department Radiation Sciences, Umeå University, Sweden FOU Unit, Östersund Hospital, 831 35 Östersund, Sweden; bertil.axelsson@regionjh.se

**Keywords:** palliative care, provision of palliative care, equal availability

## Abstract

The European Council, the World Health Organization, the International Association of Hospice and Palliative Care, and various other national guidelines emphasize equal provision of palliative care. To fulfill this vision, all involved need to be aware of the existing situation even in western European countries. Data from the European Atlas of Palliative Care and the Swedish Registry of Palliative Care are used to illustrate the present inequalities. The data illustrate the unequal provision of palliative care relating to level of care, place of residence, diagnoses, and age. The challenge of providing equal palliative care remains, even in Western European countries, in spite of all positive developments. Different approaches that may contribute to successful implementation of equal palliative care are discussed. The challenge is still there and will require some effort to resolve.

## 1. Introduction

According to the World Health Organization (WHO), more than 50 million people every year need palliative care globally and this number will increase due to aging populations and medical progress. The WHO also estimates that only 12% of those in need actually receive palliative care [1]. The European council emphasizes that the consequence of unequal provision of palliative care is unnecessary suffering for both patients and their family members and concludes that adequate availability to palliative care prevents unnecessary hospital admissions and promotes a more effective use of healthcare services [2].

Both the recommendations from the European Commission (2003) [3] and the World Health Assembly (2014) [4] state clearly that availability of palliative care according to individual need is an ethical responsibility for both governments and healthcare professionals and should be available throughout the continuum of care. Affirming access to palliative care and to essential medicines contributes to the realization that people should have the right to the highest attainable standard of health and well-being.

The consensus-based definition of palliative care presented by the International Association of Hospice and Palliative Care (IAHPC) in 2020 [5] further stresses that palliative care should be available to everyone, irrespective of age, throughout the course of an illness, according to need, in all healthcare settings (place of residence and institutions), and at all levels of care (primary to tertiary) and be an integral part of basic, intermediate, specialist, and continuing education. However, a newsletter from the WHO 2021 [1] clearly states that the largest proportion of patients with palliative care needs occur in low- and middle-income countries (78%). This statement may incorrectly be interpreted as unequal provision of palliative care in high-income countries is not an issue. Since 2007, the European Association of Palliative Care (EAPC) has published the EAPC Atlas of Palliative Care to enable comparisons between European countries and to monitor palliative care provision in Europe.

Sweden is one example of a country that is classified as a high-income country belonging to the group of countries with the highest development of palliative care [6]. Since 2005, the internationally unique Swedish Registry of Palliative Care (SRPC) has collected data about palliative care measures performed during the last week of life from all deaths regardless of diagnosis, age, and level of care [7]. The data come from a web questionnaire that includes about 30 questions answered by attending staff. Questions cover important care aspects during the last week in life, such as whether the patient suffered from pain, anxiety, nausea, dyspnea, confusion, death rattle, or pressure ulcers, whether patients and next-of-kin had received information about the transition to end-of-life care, place of death, type of palliative care interventions (specialized palliative consultations, admitted to specialized palliative unit), whether injectable PRN drugs were prescribed, and if symptoms were assessed in a structured way etc. [7]. The annual coverage has been about 60,000 registered deaths, which is approximately 66% of all deaths in the country. Even though palliative care should be involved as early as needed in any incurable disease trajectory, the strength of SRPC data is that is provides information about all deceased regardless of age, diagnosis, place of death and level of care. The SRPC provides up-to-date information about some important aspects of palliative care both via real time data available from the internet and annual reports. According to the SRPC, approximately 80% of those who die annually die an expected death (i.e., from one or more long-term incurable diseases). From this, it can be assumed that 80% of dying patients will have some need for palliative care. However, it is more difficult to ascertain the average length of palliative care needs and to approximate the annual percentage of dying patients who need specialized palliative care occasionally or permanently. Based on descriptive data from the United Kingdom [8] and Canada [9], about 25% of all deaths annually, regardless of diagnosis and age, need specialized palliative care occasionally or permanently.

In spite of Sweden being classified as a country belonging to the highest development level of palliative care, the professionals involved know that there still are white spots representing unmet palliative care needs. An example is the spotty availability to palliative care for children and teen-agers. Sweden has 10 million inhabitants spread over a substantial area with an average of 25 inhabitants per square kilometer. There is only a couple of specific pediatric palliative teams in the whole country (each of them located in a large city, just covering the nearby area). A substantial number of adult specialized palliative teams do not admit children at all and some local communities claim that home care of children is not their responsibility. Making palliative care equally available to children outside hospitals, regardless of place of residence, would require a seamless organization that covers the whole geographical area by close cooperation with local specialized palliative teams, local community staff (e.g., district nurses, auxiliary nurses, and physiotherapist), and pediatric experts.

The aim of this paper is to present available data from Sweden and other western European countries that can illustrate the existence of white spots where the provision of palliative care is not likely to cover the needs of palliative care in the population. Hopefully this may increase the awareness of the existing inequalities and stimulate further improvements in the availability of adequate palliative care.

## 2. Materials and Methods

To get an estimation of unmet palliative care needs among the population of a specific country, we have to use some kind of surrogate indicators. One such indicator is the mix of existing specialized palliative care units in a specific country. The underlying assumption is that the need of specialized palliative competence may emerge wherever a person with incurable disease is treated. Accordingly, there has to be an appropriate mix of units that are able to reach out to the person in need regardless of place of residence, level of care, age, and diagnosis. The existence of specialized palliative consultative teams is regarded as especially important as well as an appropriate mix of other specialized palliative units. The EAPC Atlas of Palliative Care in Europe 2019 [10], was used to retrieve the number of palliative care units per 100,000 inhabitants in different western European countries and the proportion of different palliative care units in these countries.

Another surrogate indicator is the percentage of all deceased persons that have received some kind of specialized palliative care. These figures were collected from the SRPC [11] and could be presented from 21 different geographical regions in Sweden. SRPC data also provide an insight into how often assessments of pain and symptoms are documented during the last week of life. Other than minor variations between different care contexts may illustrate unequal availability of palliative care. Data presented in the result section are descriptive.

## 3. Results

In western European countries, the number of palliative care units per 100,000 inhabitants is between 0.6 and 2.2. The proportions of different specialized palliative care units vary considerably between countries: hospital support teams vary from 5–6% (Finland and Sweden) to 65% (France); home care units vary from 12% (the Netherlands) to 73% (Sweden); and in-patient hospice care varies from 0–1% (Belgium, France, Iceland, and Norway) to 42% (Italy; Table 1).

In Sweden, according to SRPC data, approximately 10,000 patients were admitted to a specialized palliative care unit and 3000 received specialized palliative consultation annually (i.e., 13,000 patients per year received specialized palliative care).

According to data presented by the SRPC, systematic assessment of pain and other symptoms is not part of a working routine during the last week of life and varies substantially between geographical regions and care contexts. The national average percentage of patients who have documented symptom assessment during their last week of life is 16.7% (*n* = 13,028) in hospitals, 31.8% (*n* = 27,639) outside hospitals, and 51.2% (*n* = 10,562) in specialized palliative units, and the national average percentage of patients who have documented pain assessment during the last week of life are 30.5% (*n* = 13,028) in hospitals, 52% (*n* = 27,639) outside hospitals, and 76.5% (*n* = 10,562) in specialized palliative units.

Specialized palliative care is not available for people with complex needs in a predictable way. Depending on where one resides (any of the 21 geographical regions in Sweden), the percentage of patients who have contact with specialized palliative care (consults, home care, or hospice/specialized hospital ward) varies between 10.5% and 25.8%. If not admitted to a specialized palliative care unit, the chance of receiving specialized palliative care support via some kind of consultation service varies. The proportion of patients who received palliative support via consultation varied according to geographical region (between 2.3% and 13.8% (Table 2)).

## 4. Discussion

The results presented above illustrate some aspects of the still prevailing inequities of the provision of palliative care in western European countries, including Sweden. According to the data in Table 1, the number of palliative units per 100,000 inhabitants varies by a factor of three. This rather wide variety may be a sign of a striking variation between different countries’ populations availability to palliative care. Fewer units with higher capacities could possibly level out potential inequities, but data on this are not available. Furthermore, the mixture of specialized palliative care units varies substantially between countries. The desire to provide equal availability to palliative care based on individual need regardless of level of care and point of time in the disease trajectory prompts many questions. For example, as hospices reach a very selective patient population during a rather brief time window close to death, how can patients receive palliative care if they are not willing to be referred to a hospice, are in a much earlier phase of disease, and live in a country with a high proportion of in-patient hospices? Hospital palliative units are important for those who need hospital care to get prompt and skillful assistance for complex palliative needs, but most patients prefer to be discharged as soon as possible so they can spend as much time as possible in their own home. How can equal access to palliative care be guaranteed in countries with few palliative home care units?

A resource-effective way to supply palliative competence where it is needed is to use palliative support teams or consulting teams [13,14,15]. Table 1 reveals varying proportions of palliative hospital support teams between different countries (from 0% to 65%). These data do not reveal whether these teams’ activities extend beyond the hospital walls. A major proportion of patients needing palliative care spend most of their time outside of hospitals receiving treatment from community and primary care. To ensure availability on an equal basis, hospital-based consultation teams should be required to extend their actions outside of hospitals to support patients with complex palliative care needs at home or at nursing homes. Countries with none or few support teams will inevitably have difficulties making adequate palliative competence available to all those in need.

Compared to a proportion of approximately 25% of all dying persons needing specialized palliative care suggested by studies in the UK [8] and Canada [9], the SRPC data implies that about 9500 (11%) of incurable and dying patients in Sweden (total annual number of deaths ~90,000) do not receive the specialized palliative care they need. The existence of unequal availability to palliative care in Sweden is further underlined by the geographical differences found in proportions of patients receiving specialized palliative care (Table 2). The existence of more complex palliative needs can be assumed to be rather equally frequent in different parts of Sweden. Accordingly, a span from 10.5% to 25.8% of all deceased that received any kind of specialized palliative care support, must be interpreted as a sign of still remaining white spots in the availability to palliative care.

Further inequalities of the availability to palliative care in Sweden have been reported based on data from the SRPC. Older age decreased the probability of implementing appropriate palliative measures during the last week of life and involving a specialized palliative team [16,17]. Diagnosis seems also to be of importance as previous studies conclude that a patient who is dying of cancer is more likely to be offered palliative interventions than patients dying of amyotrophic lateral sclerosis [18], COPD [19], dementia [20], or stroke [21].

It is reasonable to assume that patients admitted to specialized palliative care units have more complex symptoms than those treated elsewhere. However, the finding of lower proportions of patients being assessed for symptoms and pain when receiving end-of-life care at hospitals (16.7% and 30.5%) compared to patients outside hospitals (31.8% and 52%), strengthens the impression of unmet palliative care needs among patients admitted to hospital.

Palliative care is a relatively new concept. Dame Cicely Saunders launched what came to be the start of modern palliative care by opening St Christopher’s hospice in 1967 south of London. The recommendations published by the European Commission in 2003 [3] was the first international document that identified the potential positive impact of palliative care as 45 signing states agreed to implement the palliative care recommendations as soon as possible. The document clearly advocates for international acceptance of palliative care:
Palliative care is a vital and an integral part of health services. […] Any person in need of palliative care should be able to access it without undue delay, in a setting which is, as far as reasonably feasible, consistent with his or her needs and preferences [and the care] must not be influenced by disease type, geographical location, socio-economic status or other such factors.[3]


The figures presented above from western Europe and Sweden signal that the implementation of palliative care is well underway, but it still has substantial ground to cover until the goal of equal availability is reached. Implementation of new treatments is always a challenge. Implementation of palliative care is even more challenging, as it represents a new paradigm. Making palliative care more available requires a complex team approach consisting of medical, nurses, psychologists, physiotherapists, occupational therapists, dieticians, and existential measures complemented by communicative skills. This complex team approach can also complement existing services in most areas of healthcare.

In a recent article [22], Abu-Odah, Molassiotis, and Liu describe the challenges facing implementation of palliative care in low- and middle-income countries, which are a mix of personal (patient), health system (staff), organizational, and policy factors. In their systematic review, they emphasize many important aspects which may be equally valid to secure equal availability to palliative care in western European countries. As the results of this study illustrate a need of further development of palliative care, it is worth reflecting on how different implementation challenges can be approached.

It is a challenge to make the general public more open to the potentially positive effects of competent palliative care. If so, they could more readily demand access to adequate palliative care and put pressure on decision-makers. The ongoing COVID-19 pandemic has had the opposite effect. Insufficient end-of-life care has been confused with palliative care. None of the fundamental components of palliative care, (e.g., symptom control, family support, and communication/relation) or a team approach have been delivered to these dying patients and their families. The fear of death and poor care keeps patients with other incurable diseases and their families from demanding referral to palliative units. After finally being admitted to palliative care, patients and their families often wonder why they did not receive this care earlier. Palliative care professionals can promote this by collaborating with journalists to share “good examples” of palliative care regardless of incurable diagnosis, including using the education opportunities provided by the internet and participating in interviews on traditional media. 

Healthcare staff, including physicians, need solid education in palliative care and medicine starting with their basic medical training. This a prerequisite to be able to detect patients with palliative care needs but also to increase the awareness of existing “white spots” where the provision of palliative care does not meet the needs of the population. As noted by EAPC Atlas, increasing numbers of nursing and medical schools offer compulsory courses in palliative care, although many do not. Too many medical schools, including most medical schools in Sweden, exert their right to decide what to teach, choices that often do not include palliative care competence. This lack of training leaves young doctors unprepared for the clinical reality they will inevitably encounter and reinforces misunderstandings about palliative care among physicians and other healthcare staff, increasing the thresholds for appropriate referral routines to and collaboration with palliative care. Compulsory courses in palliative medicine/care for everyone attending medical school and nursing school should be the minimum.

At an organizational level, it is crucial that physicians and general practitioners who care for patients with incurable diseases have enough time to perform meticulous assessments of these patients to enable proactive treatment plans and communicate these to patients, next-of-kin, and involved staff. In Sweden, all incurable patients in home care (except those admitted to specialized palliative home teams), nursing homes, and other community care facilities are supposed to be treated by general practitioners. However, little time is available to perform home visits and medical rounds, a situation that results in unproductive and avoidable hospital admissions.

To guarantee an equal availability to specialized palliative competence (i.e., when ordinary care does not manage to relieve occurring problems), an infrastructure of specialized palliative care is needed. This infrastructure should include more than referral and admittance to a specialized palliative unit as referral and admittance usually occur rather late–i.e., just before death. However, patients and their families with complex needs can be addressed much earlier using palliative consultation at the ordinary unit of care. The specialized palliative infrastructure needs to contain a consultation team (at least a physician and nurse available for community care, primary care, and at different hospital departments), specialized palliative home care units, specialized palliative hospital beds, hospice, and a specialized palliative outpatient clinic. These units need to cover the whole geographical area and be available 24/7, including a specific specialized on-call staff available regardless of diagnosis, age, and level of care.

In Sweden, the national guidelines on palliative care [23] recommend this kind of infrastructure, but legislation has not been put into place, resulting in a varying and very much ad hoc infrastructure over the country. Care delivered by hospitals and primary healthcare centers are decided by 21 independent county councils. Community care such as home care and nursing home care are governed by 290 different local municipalities, a situation that further emphasizes the need for not only national legislative measures but also allocation of appropriate financial resources to enable equal availability to palliative care. The need of both national guidelines and a secure funding by a health finance system are stressed by Clark et al. [6] as essentials for a country to reach the highest level of palliative care development.

A weakness with this study is that the results are based on indirect data. The data gathered in the European Atlas of Palliative Care are mainly based on personal communications with one or two experts per country, a data collection strategy that increases the risk of incomplete data. In addition, the more detailed information generated by the SRPC is based on staff reporting, not patient reporting, and only covers Sweden. Another limitation is that SRPC data mainly covers palliative care provision during the last week in life, thus not allowing any data on all palliative activities undertaken earlier in the disease trajectories. However, this study has relied on data meticulously collected from all European countries, which allows for relevant comparisons. In addition, the data from SRPC cover all deaths regardless of diagnosis, age, place of residence, and level of care. As all SRPC data covers exactly the same time window (i.e., the last week in life), this strengthens comparative validity.

## 5. Conclusions

Even if the development of palliative care proceeds, and in many areas it has taken substantial steps forward, the availability to palliative care is far from equal even in countries in western and northern Europe. The first step required to initiate improvements is to realize the existence of the problem. Hopefully, this article will leave it beyond doubt that the problem is real and inspire healthcare professionals, politicians, and administrators to design policies and best practices to come closer to the vision of equal availability to palliative care according to individual need regardless of diagnosis, age, place of residence, or level of care. The challenge is there to accept.

## Figures and Tables

**Table 1 ijerph-19-04229-t001:** Data from the EAPC Atlas of Palliative Care in Europe 2019 depicting numbers/100,000 inhabitants and different proportions of palliative care (PC) units in different western European countries.

Country	PC Units/100,000 Inhabitants	PC Units Home Care (%)	PC Units Hospital Unit (%)	PC Units Hospital Support Team (%)	PC Units In-Patient Hospice (%)	PC Units Mixed Units (%)
Austria	2.2	35	22	37	6	-
Belgium	1.7	14	26	59	1	-
Denmark	0.9	-	17	-	35	48
Finland	0.7	59	26	5	10	-
France	1.0	14	21	65	-	-
Germany	1.1	31	37	7	25	-
Ireland	1.9	37	2	49	12	-
Iceland	1.5	40	20	40	-	-
Italy	0.9	58	-	-	42	-
Netherlands	0.9	12	16	41	31	-
Norway	1.1	-	34	-	-	66
Portugal	0.9	22	18	46	14	-
Spain	0.6	38	24	34	1	-
Switzerland	1.4	29	41	27	3	-
Sweden	1.6	73	6	6	15	-
United Kingdom	0.9	34	-	40	26	-

**Table 2 ijerph-19-04229-t002:** Proportions of all dying patients reported to SRPC regardless of diagnosis, place of residence, and level of care who received support from a specialized palliative unit [12].

Region	Specialized Home Care (%)	Specialized Ward (Hospital/Hospice) (%)	Specialized Consultation (%)	Specialized Palliative Support (%)
Blekinge	2.9	1.2	15.5	19.6
Dalarna	7.8	1.6	3.0	12.3
Gotland	4.7	10.9	10.2	25.8
Gävleborg	2.8	0	13.8	16.6
Halland	1.4	3.8	12.6	17.9
Jämtland	5.8	1.6	12.8	20.2
Jönköping	3.5	2.8	13.6	19.9
Kalmar	1.3	6.5	7.0	14.8
Kronoberg	1.6	0.1	12.4	14.1
Norrbotten	0.4	5.2	11.9	17.5
Skåne	5.9	7.7	3.9	17.5
Stockholm	5.9	14.7	3.6	24.1
Södermanland	4.6	8.0	4.5	17.0
Uppsala	3.4	7.3	4.7	15.4
Värmland	2.4	0.7	13.4	16.5
Västerbotten	3.6	7.1	9.3	20.0
Västernorrland	4.0	3.9	2.7	10.5
Västmanland	1.9	6.4	2.3	10.6
Västra Götaland	2.0	5.7	9.4	17.1
Örebro	1.3	6.3	6.2	13.9
Östergötland	8.2	5.5	6.0	19.7
**Sweden** **(*n* = 58,522)**	**4.0**	**6.9**	**7.3**	**18.2**

## Data Availability

Data sources are found among references.

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
