# Peer review of "The Challenge: Equal Availability to Palliative Care According to Individual Need Regardless of Age, Diagnosis, Geographical Location, and Care Level"

_ijerph, 2022, doi:10.3390/ijerph19074229_

Round 1

Reviewer 1 Report

Thank you for the opportunity to review this paper. It is well written and very easy to read.  I have only a couple of suggestions for improvements and slight English language change in a few places. Good luck!

Author Response

I could not find any specific suggestions in this document, which I communicated in a mail sent to the editor 7th March. I have not got any further response from you.

The submitted paper had gone through a thorough review by a native English speaker.

Reviewer 2 Report

This is an interesting paper that discusses the need for equal availability to palliative care according to individual need regardless of age, diagnosis, geographical location, and care level. This paper would be interesting for publication but it needs major revision since it is at least for my a bit confusing what is the aim of the paper.

The paper presents Swedish situation and tries to compare to the situation in other European countries but it does in a confusing manner.

To make the paper more understandable the following is required:

  1. In the introduction section I would explain more about the Swedish Registry of Palliative Care There are 30 questions in it what areas do they cover. Has data form it already been analysed and published and by whom. Do other countries have something similar or not? Then I would come to the issues explored in the paper have they been explored before and why did the author decide to analyse them? What is the aim of this paper this research? The questions presented are about unmet needs and how are they connected with equal availability? I would concentrate on Sweden and use other countries in discussions section (The EAPC Atlas of Palliative Care in Europe 2019).

  1. In the methods section I would concentrate on the analysis of data from the Swedish registry and what statistical methods were used are those descriptive statistic methods or other methods as well.

  1. Then I would present in the results section just the data form the Swedish registry and comparison I would leave for the discussion section. I would concentrate on equal availability to palliative care according to individual need regardless of age, diagnosis, geographical location, and care level in Sweden and then in the discussion section I would address why is it so.

  1. In the discussions section I would then compare data from Swedish registry to the data The EAPC Atlas of Palliative Care in Europe 2019. Can they be compared (do they use the same methodology for data collection, do they ask different questions). And then form the data presented and the discussion I would draw conclusions.

Author Response

The aim of the paper is now clearly stated (rows 90-94).

The Swedish Registry of Palliative care is presented in more detail (55-61) including the type of data collected.

In the method section a clear rational for the use of chosen surrogate indicators is presented (97-112) and it is stated that descriptive methods are used (116).

By providing a rational for the data collected I still find that both data from the EAPC Atlas and the SRPC follow the aims of the paper and contributes to the conclusion.

The discussion has been complemented by reflections on the SRPC results (199-208).

Reviewer 3 Report

The author reviewed secondary data examining involvement of palliative care (PC) at end of life for patients in Sweden and Western Europe. This will be an important addition to the literature. At this time, the manuscript requires major revisions. 

Abstract- Will need to be updated once comments below are incorporated.

Intro- The introduction was mostly fine. It introduced the problem well enough. Do be sure that references are all cited. On page one, lines 40-41, it is cited that the WHO 2021 states that low- and  middle-income countries have the highest need. It is then stated that this implies that PC in high-income countries is not a problem. This is not true. The statement by the WHO does not imply this, it implies that the highest need is in low- and middle-income countries. It might also imply that higher-income countries have better service than low- or middle-income countries. It does not state that there is no problem.

It is stated that it is unclear how palliative care is applied in higher income countries. This is missing important literature. There is international work done in this area by

  • Lynch, T.; Connor, S.; Clark, D. Mapping Levels of Palliative Care Development: A Global Update. J. Pain Symptom Manag. 2013, 45, 1094–1106.

This citation that I provide here is not their most updated work, either.

As well, Canada (a world leader in PC and a higher-income country) has published their own data. See

  • Highlights from the National Palliative Medicine Survey. Available online: http://www.cspcp.ca/wp-content/uploads/2015/04/PM-Survey-Final-Report-EN.pdf.

As well, there are qualitative reports from providers in Canada stating what are the gaps in their PC, see

  • Robinson, M.C.; Qureshi, M.; Sinnarajah, A.; Chary, S.; de Groot, J.M.; Feldstain, A. Missing in Action: Reports of Interdisciplinary Integration in Canadian Palliative Care. Curr. Oncol. 2021, 28, 2699–2707.https://doi.org/10.3390/curroncol28040235.

It also seems that PC services you’re investigating are End of Life services. This is only part of the PC scope of care. Perhaps address this in your introduction. For example, talk about full scope and then narrow down the piece of care you’ll discuss in your article.

On page 2, lines 61-72, there a number of questions that “can be asked.” Are these the questions that were asked? If so, they belong in the methods section. If not, I’m confused as to their relevance.

Materials and Methods- Please structure this section with the necessary sections. As a reader, I do not know what was done. What was the procedure? What date range was used? What were the inclusion and exclusion criteria? What statistical software was used? What about missing data?

Results- It seems that the results are inconsistently reported about Western Europe or about Sweden. I am unsure what your database contained. Perhaps it would be helpful to structure results by Western Europe and Sweden with specific subheadings?

On page 3, line 102, there are references to Canadian and UK data. Perhaps include the relevant stats from these publications so that readers can easily see the comparison being made?

Of note, patients have incurable disease as opposed to being incurable patients.

Patients within their last week of life is a very late introduction of palliative care. Please include reasons for using this benchmark. Or, perhaps in Sweden/Western Europe, PC is not fully developed and this is all that’s offered? As opposed to other countries or continents where optimal palliative care is integrated early? See

  • Temel JS, Greer JA, Muzikansky A, Gallagher ER, Admane S, Jackson VA, Dahlin CM, Blinderman CD, Jacobsen J, Pirl WF, Billings JA. Early palliative care for patients with metastatic non–small-cell lung cancer. New England Journal of Medicine. 2010 Aug 19;363(8):733-42.
  • Charalambous, H., Pallis, A., Hasan, B. et al.Attitudes and referral patterns of lung cancer specialists in Europe to Specialized Palliative Care (SPC) and the practice of Early Palliative Care (EPC). BMC Palliat Care 13, 59 (2014). https://doi.org/10.1186/1472-684X-13-59
  • Wentlandt K, Krzyzanowska MK, Swami N, Rodin GM, Le LW, Zimmermann C: Referral practices of oncologists to specialized palliative care. J Clin Oncol. 2012, doi:10.1200/JCO.2012.44.0248

It seems that there was also qualitative data collected (Page 4, lines 129-131)? This is a surprise. Where did this come from?

I suspect that page 4, lines 129-137 are better for the discussion than the results.

Discussion- Although I tend to agree with some of the suppositions included in the discussion, I would argue that the discussion is currently being used as an editorial. It needs to stick closer to the findings and offer interpretations of these, rather than solutions for the findings that haven’t been clearly drawn out for the reader. Also, be sure to be using your data to identify problems and/or cite references used.

Page 5, lines 164-174, this belongs in the intro.

Conclusion- Similar comment to the discussion- be sure that these follow from the data collected and interpreted herein.

Limitations- From what I know about this study, these seem reasonable.

Future Directions- Perhaps include further research ideas that stem from what was found herein.

Author Response

The abstract is still valid.

The wording regarding high income countries has been modified (row 44).

The most recent study by Clark et al, about global palliative development, has been added as a reference (reference 6, rows 51 and 300)

That palliative care covers more than end-of-life care is expressed both in the introduction (62-63) and when discussing limitations (308-310).

The questions on page 2 has been deleted as their relevance to this study were not obvious.

In material and methods the rational for choosing the presented variables has been stated (97-115).

The content of the SRPC data base is exemplified in the introduction (56-65)

The wording “incurable patients” have been changed to the more appropriate “patients with incurable disease” (271).

That palliative care is provided and needed earlier than at the end-of-life is expressed both in the introduction (62-63) and in limitations (308-310). The rational behind the chosen surrogate indicators is presented in material and methods (97-115) and does not imply that western European countries do not offer any palliative care earlier in the disease trajectories.

In the introduction the nature of SRPC data is exemplified (55-61), which will help the reader to understand the origin of presented results.

Lines have been moved to the discussion as proposed (199-201)

That the proportion of patients assessed for symptoms and pain in different care context also is relevant to illustrate unequal provision of palliative care, is now declared in the material and methods section (114-115) and also stressed in the discussion (211-216).

The discussion has been revised to more clearly comment on the results and why suggested measures can contribute to further improvements.

Round 2

Reviewer 2 Report

The paper is now much improved still there is additional work to be done on the manuscript. In the results section there are still parts that should be put in discussion. The results section gives an overviews of the results not the discussion of other studies. For example in the line 158 to 160. Also the text from line 197-203. The authors should describe the results in tables in the results section and comparison with other Swedish and  UK studies in the discussion section.

Author Response

As suggested I have removed the two lines 135-137 as the message already was made clear in the discussion (lines 202-204).

As also suggested I have moved the lines 165-170 to the discussion.

It was confusing to me that the lines mentioned both by reviewer 2 and the editor did not make sense, but I hope that the made revisions comply with your intentions.

I still prefer to present data on lines 147-151 only in text.